# Morphological Characteristics and Comparative Chloroplast Genome Analyses between Red and White Flower Phenotypes of *Pyracantha fortuneana* (Maxim.) Li (Rosaceae), with Implications for Taxonomy and Phylogeny

**DOI:** 10.3390/genes13122404

**Published:** 2022-12-18

**Authors:** Shi-Xiong Ding, Jia-Chen Li, Ke Hu, Zi-Jian Huang, Rui-Sen Lu

**Affiliations:** 1Jiangsu Key Laboratory for the Research and Utilization of Plant Resources, Institute of Botany, Jiangsu Province and Chinese Academy of Sciences (Nanjing Botanical Garden Memorial Sun Yat-Sen), Nanjing 210014, China; 2Core Botanical Gardens/Wuhan Botanical Garden, Chinese Academy of Sciences, Wuhan 430074, China; 3University of Chinese Academy of Sciences, Beijing 100049, China; 4College of Life Sciences, Zhejiang University, Hangzhou 310058, China

**Keywords:** *Pyracantha fortuneana*, red flower phenotype, chloroplast genome, comparative analysis, taxonomic investigation

## Abstract

*Pyracantha fortuneana* (Maxim.) Li (Rosaceae), commonly known as Chinese firethorn, is an evergreen shrub with high nutritional, medicinal, and horticultural importance. This species typically has white flowers, but a rare red flower phenotype has been found in very few wild populations in western Hubei, China, showing great ornamental potential. In this study, the complete chloroplast genome of the red flower phenotype of *P. fortuneana* was reported for the first time, using high-throughput sequencing technology. The complete chloroplast genome was 160,361 bp in length and showed a typical quadripartite structure with a pair of inverted repeat (IR) regions (26,350 bp) separated by a large single-copy (LSC) region (88,316 bp) and a small single-copy (SSC) region (19,345 bp). A total of 131 functional genes were annotated in this chloroplast genome, including 86 protein-coding genes (PCGs), eight rRNA genes, and 37 tRNA genes. Comparative chloroplast genome analyses revealed that high genome similarity existed not only between red and white flower phenotypes of *P. fortuneana*, but also among *Pyracantha* species. No evidence for positive selection was found in any PCG, suggesting the evolutionary conservation of *Pyracantha* chloroplast genomes. Furthermore, four mutational hotspots (*trnG-trnR-atpA*, *psbZ-trnG-trnfM-rps14*, *ycf3-trnS-rps4*, and *ndhF-rpl32*) with π > 0.004 were identified as potential molecular markers for *Pyracantha* species. Phylogenomic analysis strongly supported that the red flower phenotype of *P. fortuneana* was nested within the common white flower phenotype. Based on both morphological and molecular evidence, we suggest that the red flower phenotype of *P. fortuneana* could be considered as a new forma. Overall, the availability of these genetic resources will not only offer valuable information for further studies on molecular taxonomy, phylogeny, and population genetics of *Pyracantha* species but also could be used as potential genetic resources for Chinese firethorn breeding.

## 1. Introduction

*Pyracantha* M. Roem. (Rosaceae), commonly called firethorn, is composed of 10 evergreen shrub species and mainly distributed in contiguous areas from Eastern Asia to Southern Europe [1,2]. China is the most important distribution center of this genus and harbors seven species, including five endemic species [1,3]. Previous morphological and molecular phylogenetic analyses placed this genus in the subfamily Maloideae [3], or treated it as a subtribe (Pyracanthinae) within the tribe Maleae of subfamily Amygdaloideae [4,5,6]. However, *Pyracantha* is currently recognized as a stable monophyletic genus [7,8], differing from many paraphyletic taxa in the tribe Maleae, such as *Stranvaesia* Lindl., *Sorbus* L., *Photinia* Lindl., etc., which have reticulate evolutionary histories with undefined taxonomic positions [9,10,11,12].

*Pyracantha fortuneana* is one of the most economically important species in this genus and has great nutritional, medicinal, and horticultural properties [13]. Specifically, the fruits of *P. fortuneana*, rich in sugar (149.40 mg/g), proteins (105.00 mg/g), and vitamin C (0.32 mg/g), were traditionally consumed as meal replacements by local people after being ground into flour [13,14,15]. The roots, leaves, flowers, and fruits can be used as traditional Chinese medicine with various pharmacological activities, including antioxidative, immune, and anti-tumor effects [13,16]. More importantly, this species has a high ornamental value, with flourishing branches and leaves, and dense flowers and red fruits, and is now widely used as a wild evergreen shrub in China [14]. Typically, the flowers of this species are white or yellowish, while a red flower phenotype was found in very few wild populations in western Hubei, China. The petals and inner sepals of this phenotype are red (versus white in the common phenotype), and all mature leaves turn red (versus green in the common phenotype) in winter (Figure 1) [17]. Generally speaking, this rare red flower phenotype shows a higher ornamental value than the common white flower phenotype of *P. fortuneana* and thus could be served as a suitable parent for breeding hybrids and subsequent development of improved cultivars. However, so far, genetic and/or genomic resources for the red flower phenotype of *P. fortuneana* have not been developed, and the taxonomic status of this rare phenotype has not been verified.

Chloroplasts are believed to derive from a single primary endosymbiotic event involving the capture of a cyanobacterium and have their own genomes encoding many key proteins in relation to photosynthesis and other major cellular functions, including synthesis of starch, fatty acids, pigments, and amino acids [18,19,20]. A typical chloroplast genome is a double-stranded circular DNA and portrays a quadripartite structure, including a pair of inverted repeat (IR) regions separated by a large single-copy (LSC) region and a small single-copy (SSC) region [21]. Due to its independent matrilineal inheritance, the lack of genetic recombination, low levels of nucleotide substitution, and small effective population size, the chloroplast genome has been widely used for accurate species identification and phylogenetic inference in different plant lineages, especially those with a complex phenotypic evolution [22,23,24,25,26].

According to our field observation, we hypothesized that the red flower phenotype of *P. fortuneana* could be recognized as a rare phenotype resource of this species or be considered as a new forma. To test this hypothesis, the complete chloroplast genome of the red flower phenotype of *P. fortuneana* (*P. fortuneana* (red)) was sequenced and assembled. Combined with previously published chloroplast genomes of this genus, including two individuals of *P. fortuneana* with white flower phenotype (*P. fortuneana*-1 (white) (NC_059101) and *P. fortuneana*-2 (white) (MW596361)) and one individual each for *Pyracantha atalantioides* (Hance) Stapf (MW801001), *Pyracantha coccinea* M. Roem. (NC_062343), and *Pyracantha angustifolia* (Franch.) Schneid. (KY419957), respectively, we provided a total of six chloroplast genomes for comparative genomic and phylogenomic analyses. Our study aims were to: (1) characterize and compare the chloroplast genomes of *Pyracantha* species to demonstrate their evolutionary patterns; (2) screen and identify candidate DNA barcodes for species/phenotype identification within *Pyracantha*; (3) resolve phylogenetic relationships within the genus *Pyracantha*, and (4) gain insights into the taxonomy of the red flower phenotype of *P. fortuneana* based on both molecular and morphological evidence.

## 2. Materials and Methods

### 2.1. Sample Collection and Genome Sequencing

Young leaf materials of *P. fortuneana* with the red flower phenotype were collected from one wild population in western Hubei, China (29°21′15.68″ N, 108°58′16.61″ E, Alt. 750 m) and then dried with silica gel. Total genomic DNA was extracted from silica gel-dried leaf samples using a modified procedure of CTAB (cetyltrimethylammonium bromide) [27]. The purified genomic DNA was fragmented to construct short-insert libraries for low-depth whole-genome sequencing, using the Illumina paired-end technology platform (HiSeq-PE150), and about 8 Gb of raw data were obtained. Library construction and sequencing were conducted by Novogene Bioinformatics Technology Co., Ltd. (Beijing, China).

### 2.2. Chloroplast Genome Assembly and Annotation

The raw reads were employed to assemble the complete chloroplast genome sequence of *P. fortuneana* (red) using a GetOrganelle pipeline [28] with the suggested default parameters. The connection and circularity of the assembly graphs from GetOrganelle were subsequently visually checked in Bandage v.0.8.1 [29]. The Chloroplast genome annotation was performed using the Plastid Genome Annotator (PGA) program [30], with the annotated sequences of *Amborella trichopoda* (AJ506156) and *P. fortuneana* (NC_059101) as references. The draft annotation was further verified by GeSeq software v.1.4.2 [31] and checked manually. The annotated chloroplast genome sequence of *P. fortuneana* was deposited in GenBank (accession No.: OM793283). The circular chloroplast genomic map was drawn using the online software Chloroplot (https://irscope.shinyapps.io/Chloroplot/, accessed on 2 December 2022) [32], with subsequent manual editing.

### 2.3. Comparative Chloroplast Genome Analyses

The basic features of the chloroplast genome sequence of *P. fortuneana* (red), including the size and GC content of different regions, and gene classification were analyzed with Geneious software v.10.2.3 [33] and were compared with those in four other complete chloroplast genomes of *Pyracantha*, i.e., *P. fortuneana*-1 (white), *P. fortuneana*-2 (white), *P. coccinea*, and *P. atalantioides* (note: the chloroplast genome of *P. angustifolia* was not included in this analysis because only one copy of the IR region was kept by the GenBank submitter). The inverted repeat-single copy (IR/SC) junction characteristics of these five *Pyracantha* chloroplast genome sequences were drawn in Adobe Illustrator. Furthermore, all six chloroplast genome sequences (only one copy of IR included) from four *Pyracantha* species (i.e., *P. fortuneana*, *P. coccinea*, *P. atalantioides,* and *P. angustifolia*) were aligned to identify potentially sequence rearrangements of *Pyracantha* chloroplast genomes using Mauve software v.2.3.1 [34]. The genome-wide similarity of these *Pyracantha* species was also plotted using the online software Circoletto (http://tools.bat.infspire.org/circoletto/, accessed on 12 July 2022) [35].

### 2.4. Codon Usage and RNA Editing Analyses

For the codon usage bias analysis, MEGA v.7.0 [36] was used to calculate the RSCU (relative synonymous codon usage) values of coding sequences (CDSs) across all six *Pyracantha* chloroplast genomes. Additionally, the potential RNA editing sites in *P. fortuneana* chloroplast genomes were further predicted using the PREP-Cp web server (http://prep.unl.edu/cgi-bin/cp-input.pl) [37], with a cutoff value of 1.

### 2.5. Selection Pressure and Analysis

The selection pressure on protein-coding genes (PCGs) of six *Pyracantha* chloroplast genomes was evaluated using the Datamonkey web server (https://www.datamonkey.org/, accessed on 1 July 2022) [38], with FEL (Fixed Effects Likelihood) as the best-fit method according to the step tips. All 78 shared PCGs of six *Pyracantha* chloroplast genomes were extracted in Geneious software v.10.2.3 and aligned using the program Muscle in MEGA software v.7.0 [39]. For each gene alignment matrix, the stop codons and unaligned fragments were removed using the program Gbloks v.0.91.b [40]. Then, all these single-gene alignment matrices were concatenated into a supermatrix alignment in PhyloSuite v.1.2.2 [41]. Finally, the concatenated protein-coding sequences were uploaded to Datamonkey to perform selection pressure analysis.

### 2.6. Identification of SSRs and Highly Variable Regions

The MISA Perl program [42] was used to identify simple sequence repeats (SSRs) across chloroplast genome sequences of three *P. fortuneana* accessions (one accession of the red flower phenotype and two accessions of the white flower phenotype) and another three *Pyracantha* species (i.e., *P. coccinea*, *P. atalantioides*, and *P. angustifolia*) with the common minimum repeat settings: ten for mononucleotides, five for dinucleotides, four for trinucleotides, and three for tetranucleotides, pentanucleotides, and hexanucleotides, respectively. DnaSP software v.6.12.3 [43] was used to calculate genome-wide nucleotide diversity (Pi) of the aligned chloroplast genome sequences of *Pyracantha* (excluding one copy of IR), with a window length of 600 bp and a step size of 200 bp.

### 2.7. Genetic Distance Analyses

Pairwise genetic distances between all *Pyracantha* individuals were computed under the analysis of molecular variance (AMOVA) and used to construct a neighbor-joining phenogram [44] in MEGA v.7.0 [36] based on the multiple alignment of *Pyracantha* chloroplast genomes with MAFFT software v.7.409 [45].

### 2.8. Phylogenetic Analyses

To explore phylogenetic relationships within the genus *Pyracantha* and among the members of the tribe Maleae of Rosaceae, a total of 31 chloroplast genome sequences (only one copy of IR kept) were used to reconstruct phylogenetic trees based on the methods of maximum likelihood (ML) and Bayesian inference (BI). Except for *P. fortuneana* (red), the other 30 chloroplast genome sequences were downloaded from the NCBI database (five accessions of the subtribe Pyracanthinae, five accessions of the subtribe Vauqueliniinae, five accessions of the subtribe Lindleyinae, 13 accessions of the subtribe Mespilinae, and two outgroups (*Gillenia stipulata* and *Gillenia trifoliata*)) (Appendix A). All 31 chloroplast genome sequences were aligned using MAFFT software v.7.409 [45], and then the best-fit models for the ML and BI methods were selected according to the Bayesian information criterion (BIC) using the ModelFinder program [46] in Phylosuite v.1.2.2 [41]. ML analysis was implemented in IQ-TREE v.2.1.2 [47] with 1000 bootstrap replications under the best fit model TVM + I + G2 + F. The BI tree was constructed using MrBayes software v.3.2.6 [48], under the best-fit model GTR + F + I + G4. The Markov chain Monte Carlo (MCMC) algorithm was run for two independent runs of 1 × 10^6^ generations, with four independent Markov chain Monte Carlo (MCMC) chains each (i.e., one cold and three heated) and a sampling frequency of 1000 trees. The initial 25% of sampled trees were discarded as burn-in. The ML and BI trees were then combined in TreeGraph software v.2 [49], based on the consistent topological structures, and the combined phylogenetic tree was visualized using Figtree software v.1.4.4 (http://tree.bio.ed.ac.uk/software/figtree/, accessed on 31 July 2022).

A total of 30 nrDNA ITS sequences from two flower phenotypes of *P. fortuneana* and related species were also used for constructing phylogenetic trees following the same described methods as above for chloroplast genomes. Among these sequences, the ITS sequences of *P. fortuneana* (red), *P. fortuneana*-2 (white) and *P. coccinea* were generated from raw sequence data (NCBI SRA accession nos. SRR21976475, SRR17631715, and SRR13004386 for these three accessions, respectively) using GetOrganelle v.1.7.4 [28], while the other 27 ITS sequences (*P. fortuneana*-1 (white), four accessions of the subtribe Vauqueliniinae, four accessions of the subtribe Lindleyinae, 16 accessions of the subtribe Mespilinae, and two outgroups (*G. stipulata* and *G. trifoliata*)) were downloaded from the NCBI database (see Appendix A).

## 3. Results

### 3.1. Chloroplast Genome Characteristics

The complete chloroplast genome of *P. fortuneana* (red) was 160,361 bp in length and retained the typical quadripartite structure, comprising a large single-copy (LSC) region of 88,316 bp, a small single-copy (SSC) region of 19,345 bp, and a pair of inverted repeat (IR) regions of 26,350 bp (Figure 2). The overall GC content of the whole genome was 36.5%, and the corresponding values of the LSC, SSC, and IR regions were 34.1%, 30.4%, and 42.7%, respectively. Among the five *Pyracantha* chloroplast genomes, i.e., *P. fortuneana* (red), *P. fortuneana*-1 (white), *P. fortuneana*-2 (white), *P. coccinea*, and *P. atalantioides*, whole-chloroplast-genome sizes varied from 160,361 bp (*P. fortuneana* (red)) to 160,803 bp (*P. atalantioides*), with LSC from 88,316 bp (*P. fortuneana* (red)) to 88,698 bp (*P. atalantioides*), SSC from 19,344 bp (*P. fortuneana*-2 (white)) to 19,438 bp (*P. coccinea*), and IR from 26,342 bp (*P. coccinea*) to 26,355 bp (*P. fortuneana*-1 (white)) (Table 1). The length variations among the three individuals (two flower phenotypes) of *P. fortuneana* were much less than those among the three *Pyracantha* species. Furthermore, GC content in the LSC, SSC, and IR regions as well as whole genome sequences were almost the same among these five *Pyracantha* chloroplast genomes (Table 1).

A total of 131 functional genes, including 86 protein-coding genes (PCGs), 37 tRNA genes, and eight rRNA genes were annotated in the chloroplast genome of *P. fortuneana* (red), which could be further divided into four categories (Table 2). Among them, 18 genes were duplicated in IR regions, including seven PCGs (*ndhB*, *ycf2*, *ycf15*, *rpl2*, *rpl23*, *rps7*, and *rps12*), seven tRNAs (*trnA*-UGC, *trnI*-CAU, *trnI*-GAU, *trnL*-CAA, *trnN*-GUU, *trnR*-ACG, and *trnV*-GAC), and four rRNAs (*rrn4.5*, *rrn5*, *rrn16,* and *rrn23*) (Table 2). Ten PCGs (*atpF*, *ndhA*, *ndhB*, *petB*, *petD*, *rpl2*, *rpl16*, *rps12*, *rps16*, and *rpoC1*) and six tRNAs (*trnA-UGC*, *trnG-UCC*, *trnI-GAU*, *trnK-UUU*, *trnL-UAA*, and *trnV-UAC*) contained a single intron, while two PCGs (*clpP* and *ycf3*) had two introns (Table 2). Additionally, the *infA* gene has been entirely lost in the chloroplast of *P. fortuneana* (red), and the start codon of the *psbL* gene has mutated to noncanonical ACG instead of the most common AUG. The gene number and gene content in the chloroplast genome of *P. fortuneana* (red) were totally identical with those in *P. fortuneana* (white), *P. coccinea*, and *P. atalantioides*.

### 3.2. Whole-Chloroplast-Genome Comparison

The IR/SC junction characteristics of chloroplast genomes were almost the same not only between *P. fortuneana* (red) and *P. fortuneana* (white) but also even between *P. fortuneana* and three other *Pyracantha* species (Figure 3). The *rps19* gene of these five chloroplast genomes extended 120 bp into the IRb region at the junction of the LSC/IRb (JLB), creating a duplicated pseudogene at the IRa region (pseudogene not shown). Similarly, the *ycf1* gene crossed the SSC/IRa (JSA), and the pseudogene fragment was located at the IRb region with 1074 bp (pseudogene not shown). The gene *trnH-GUG* was completely located in the LSC region, with a length of 120–205 bp away from the LSC/IRa (JLA) boundary (Figure 3). Additionally, structural comparison of *Pyracantha* chloroplast genomes revealed that there were high levels of syntenic similarity between *P. fortuneana* (including two flower phenotypes) and three other *Pyracantha* species (Appendix A), with no significant rearrangements detected (Appendix A).

### 3.3. Codon Usage Bias and RNA Editing Sites

A total of 26,292 codons were identified in the chloroplast genome of *P. fortuneana* (red) (Appendix A). Among all amino acids, tryptophan and methionine were the only two amino acids translated by one codon (UGG and AUG, respectively), while the remaining amino acids were translated by two to six codons. The most frequently used codon was UUA (1.93%, leucine), while the least abundant codon was AGC (0.38%, serine). Meanwhile, the three most frequent amino acids were serine (6.00%), arginine (6.00%), and leucine (5.99%), whereas the two least frequent amino acids were methionine (1.00%) and tryptophan (1.00%) (Table 3). It is also important to note that nearly all (30/32) C/G-ending codons had RSCU values lower than 1, while nearly all (29/32) A/U-ending codons had RSCU values higher than 1, indicating that most of the amino acids tended to use A/U-ending codons rather than C/G-ending codons (Table 3). Among all six *Pyracantha* chloroplast genomes, the total number of codons ranged from 26,288 (*P. atalantioides*) to 26,317 (*P. angustifolia*), with a slight difference between three accessions of *P. fortuneana* and two other *Pyracantha* species (Appendix A). The RSCU values of the same codons were highly identical in six *Pyracantha* chloroplast genomes, and for each kind of amino acid, the sum of RSCU values of all codons involved in its encoding was also equal (Figure 4).

The three *P. fortuneana* chloroplast genomes (one accession with the red flower phenotype and two accessions with the white flower phenotype) shared the same 38 potential RNA editing sites, which were distributed on 17 PCGs associated with NADH-dehydrogenase (*ndhB*, *ndhD*, and *ndhF*), ATP synthase (*atpA*, *atpB*, and *atpI*), photosystem II (*psbE, psbF*, and *psbL*), the small subunit of ribosome (*rps2* and *rps14*), RNA polymerase subunits (*rpoA* and *rpoB*), subunits of cytochrome (*petB*), and other functional genes (*clpP*, *accD*, and *matK*) (Figure 5, Appendix A). Among these genes, the gene *ndhB* had the highest number of RNA editing sites (10 sites), followed by genes *ndhD* (four sites), *ndhF* (three sites), *rpoB* (three sites), *atpA* (three sites), *accD* (two sites), *petB* (two sites), and *rps14* (two sites), while all the others had only one RNA editing site (Figure 5). The conversion of these sites was all from “C” to “T”, and most of the converted sites occurred in the second base of the codon, accounting for 76.31% of all sites (29/38). Of 38 RNA editing sites, the most frequent conversion of RNA editing sites was from TCA (serine) to TTA (leucine) (13), followed by CCA (proline) to CTA (leucine) (6), CTT (serine) to TCA (phenylalanine) (4), CAT (histidine) to TAT (tyrosine) (3) (Appendix A). Correspondingly, the most frequent conversion of amino acids was from serine to leucine (15), followed by proline to leucine (7), serine to phenylalanine (5), and histidine to tyrosine (4) (Appendix A).

### 3.4. Selection Pressure

The selection pressure analysis revealed that no CDSs across all six *Pyracantha* chloroplast genomes experienced positive selection (dN/dS ratio > 1), while at least 10 CDSs, mainly involved in NADH-dehydrogenase (*ndhD*, *ndhF*, and *ndhH*), ATP synthase (*atpI*), photosystem II (*psbM*), RNA polymerase subunits (*rpoC1*), small subunit of ribosome (*rps18*), and other functional genes (*ccsA*, *matK*, and *rbcL*) had at least one site under purifying selection (Table 4), thus presumably had conserved function. In addition, a total of 42 neutral selected sites (dN/dS ratio = 1) were identified in 27 CDSs, most of which were involved in NADH-dehydrogenase (five genes, 12 sites), RNA polymerase subunits (two genes, three sites), the small subunit of ribosome (four genes, four sites), the large subunit of ribosome (three genes, three sites), photosystem II (three genes, three sites), subunits of cytochrome (two genes, three sites), photosystem I (one genes, three sites), ATP synthase (one gene, two sites) (Appendix A).

### 3.5. SSRs and Highly Divergent Regions

A total of 101 SSRs were identified in the chloroplast genome of *P. fortuneana* (red), including 68 mononucleotides (67.33%), 25 dinucleotides (24.75%), seven tetranucleotides (6.93%), and one pentanucleotide (0.99%) (Figure 6A, Appendix A). The most common motifs found in this chloroplast genome were A/T (95.6%) for mono-, AT/AT (100%) for dinucleotides, while pentanucleotides and hexanucleotides were rarely observed in the chloroplast genome of *P. fortuneana* (red) (Figure 6A,B). Within *P. fortuneana*, *P. fortuneana*-2 (white) harbored more mononucleotides and dinucleotides than *P. fortuneana* (red) and *P. fortuneana*-1 (white) in the LSC regions (Figure 6A–C). At the inter-species level, *P. atalantioides* contained the most SSRs, followed by *P. fortuneana*-2 (white), *P. coccinea*, *P. fortuneana*-1 (white), and *P. fortuneana* (red), while *P. angustifolia* contained the least (Figure 6A–C, Appendix A).

The nucleotide diversity (Pi) for each sliding window was calculated to detect highly variable regions of *Pyracantha* chloroplast genomes. The Pi values for each window varied from 0 to 0.01267, with an average value of 0.00068. The LSC (average Pi = 0.00077) and SSC (average Pi = 0.00108) regions displayed higher sequence divergences than the IR (average Pi = 0.00008) regions (Figure 7). A total of four highly variable regions, i.e., *trnG-trnR-atpA*, *psbZ-trnG-trnfM-rps14*, *ycf3-trnS-rps4*, and *ndhF-rpl32* with Pi > 0.004 were detected (Figure 7), which could be used as DNA barcodes in this genus. Among these highly variable regions, *ycf3-trnS-rps4* can be expected to successfully distinguish individuals with the red flower phenotype from those with the white flower phenotype.

### 3.6. Genetic Distance

AMOVA analysis revealed that there was almost no genetic difference among three *P. fortuneana* individuals (two flower color phenotype) and between *Pyracantha* two species (*P. fortuneana* and *P. angustifolia*). The genetic differences mainly occurred between *P. coccinea* and the rest of five *Pyracantha* individuals (Table 5). Furthermore, the visualized neighbor-joining tree also showed that the genetic distances were short among three *P. fortuneana* individuals and between *P. fortuneana* and *P. angustifolia,* while longer genetic distances were observed between *P. coccinea* and five other *Pyracantha* individuals (Figure 8).

### 3.7. Phylogenetic Relationships

The phylogenetic tree based on chloroplast genome sequences strongly supported that the genus *Pyracantha* was monophyletic and was also the only genus in the subtribe Pyracanthinae. This genus was further recovered as sister to the subtribe Mespilinae, comprising the genera *Crataegus*, *Hesperomeles*, *Amelanchier*, and *Malacomeles*. Within *Pyracantha*, the accession with the red flower phenotype of *P. fortuneana* (*P. fortuneana* (red)) was nested within the accessions with the white flower phenotype (*P. fortuneana* (white)), rather than formed a sister relationship to *P. fortuneana* (white) (Figure 9). Phylogenetic relationships within the genus *Pyracantha* and among the members of the tribe Maleae of Rosaceae inferred from ITS sequences (Appendix A) were highly similar to those inferred from chloroplast genome sequences.

## 4. Discussion

### 4.1. Chloroplast Genome Features

Chloroplast genomes within a species are highly conserved in terms of genomic structure, gene content, gene order, and GC content [50,51], with *P. fortuneana* being no exception in this regard. All three individuals of *P. fortuneana* (two flower phenotypes) possessed the typical quadripartite structure of land plant chloroplast genomes, with a pair of IR regions (26,350–26,355 bp) separating LSC (88,316–88,393 bp) and SSC (19,344–19,348 bp) regions, and encoded 113 identical unique genes, including 79 PCGs, 30 tRNAs, and four rRNAs (Figure 2, Table 1 and Table 2). All three chloroplast genomes also shared the same overall GC content (36.5%), higher than that in LSC (34.1%) and SSC (30.4%) regions, but lower than that in IR regions (42.7%) (Table 1), likely due to the high GC content of the four rRNAs. The high GC content in IR regions may also contribute to the stability of their chloroplast genomes [52,53,54]. Moreover, the location of the IR/SC boundaries were nearly identical either within *P. fortuneana* or in comparison to *P. atalantioides*, *P. coccinea*, and *P. angustifolia*, and no gene arrangements were detected in *Pyracantha* chloroplast genomes (Figure 3 and Appendix A). At a broader taxonomic scale, the genome size, GC content, and gene number of *Pyracantha* chloroplast genomes also resembled those of previously published Maleae species [55,56]. As an example, the gene for the translation initiation factor, *infA*, which was found to be lost in *Pyracantha* chloroplast genomes, was also widely absent in the chloroplast genomes of the tribe Maleae and Fabaceae and may have been transferred to the nuclear genome or replaced with other related genes [57].

RNA editing is a post-transcriptional process on the target transcripts by base insertion, deletion, or replacement [58]. It mainly occurs in chloroplast and mitochondrial genomes, and the number of editing sites varies in terrestrial plants [59]. In this study, however, the RNA editing sites and codon usage bias were completely identical not only between red and white flower phenotypes of *P. fortuneana* but also even among *Pyracantha* species (Figure 4 and Figure 5), strongly supporting the functional conservation of RNA editing and translation in *Pyracantha* chloroplast genomes. In addition, the ratio of nonsynonymous to synonymous substitutions (dN/dS) has been widely applied to evaluate the selection pressure sites and nucleotide evolution rates in coding sequences [60]. It is worth noting that only purifying and neutral selected sites were detected in *Pyracantha* chloroplast genomes (Table 4 and Appendix A), indicating the conserved functions of these chloroplast genes in their evolutionary history [61]. Together, these findings provided further evidence on the conserved nature of *Pyracantha* chloroplast genomes.

Comparative chloroplast genome analyses revealed that most of the sequence variations were found in the LSC and SSC regions, while the IR regions exhibited comparatively fewer sequence variations (Figure 7). The lower sequence divergence was observed in the IRs than the LSC and SSC regions, which may be due to copy correction between IR sequences by gene conversion [61]. Furthermore, a total of four highly variable regions, i.e., *trnG-trnR-atpA*, *psbZ-trnG-trnfM-rps14*, *ycf3-trnS-rps4*, and *ndhF-rpl32* (Pi > 0.004) (Figure 7) identified in this study could be served as molecular markers for future phylogenetic, population genetic and phylogeographic studies of *Pyracantha*.

### 4.2. Morphology, Phylogeny, and Taxonomy of the Red Flower Phenotype of P. fortuneana

Morphologically, the red flower phenotype of *P. fortuneana* is evergreen or semi-evergreen shrubs (mature leaves turn red in winter), up to 3 m tall. Lateral branches are short, and their apex is thornlike. Young branchlets are densely rusty pubescent, and mature branchlets are dark brown and glabrous. Buds are small and cover pubescent in the outer. Leaves mainly grow on the short branches, and petioles are glabrous or slightly pubescent when young. Leaf blades are obovate or obovate-oblong and glabrous in both surfaces, 1.5–5.5 cm long, 0.5–2 cm wide. Leaf base is cuneate to wide round, and the serrations of leaf margin are conspicuous or inconspicuous. Leaf apex is obtuse or emarginate. Dense flowers are clustered into loose compound corymbs, ca. 25 cm in diameter. Peduncles are nearly glabrous and caducous bracts are lanceolate. Pedicels are nearly glabrous, ca. 1 cm long. The color of flowers is red to pink, ca. 1 cm in diameter. Calyx tubes are campanulate and their outer surfaces are glabrous. Sepals are triangular to triangular-ovate and glabrous, 1–1.5 mm long, with entire margin and blunt apex. Petals are red to pink and nearly round, ca. 4 × 3 mm long, and apex is rounded or blunt. Stamens are 20 and filaments are red to pink, 3–4 mm long. Ovaries are densely white pubescent on the upper part and styles are red to pink, 3–4 mm long. Pome is orange-red to dark red and subglobose, ca. 5 mm long. Fruit pedicels are short, 2–5 mm long. Sepals are persistent in fruit apex and erect. The florescence is from April to May, and the fruiting period is from August to November [1,3].

Although *P. fortuneana* individuals with the red flower phenotype can be conspicuously distinguished from those with the white flower phenotype, by their pink to red floral parts, including inner sepals, petals, styles and filaments, and red mature leaves (Figure 1, Table 6), most vegetative morphological traits between red and white flower phenotype individuals are always the same. For example, both of them have short, thornlike lateral branches, rusty-pubescent young branchlets, dark brown and glabrous mature branchlets, obovate or obovate-oblong leaf blades, short, glabrous and pubescent petioles [1,3,13]. Although Wang [17] proposed that *P. fortuneana* individuals with the red flower phenotype appear to have a wide round leaf base and conspicuous serrations, different from the characteristics of cuneate leaf base and inconspicuous serrations in individuals with the white flower phenotype, our field observations showed that the red flower phenotype individuals also harbored the above traits in white flower phenotype individuals (Figure 1). Thus, except for flower color, individuals with these two flower phenotypes are not clearly distinguished. Furthermore, in terms of geographical distribution, *P. fortuneana* individuals with the red flower phenotypes were only found in the Enshi Tujia and Miao Autonomous Prefecture, western Hubei, China, and have a scatted distribution in thickets, stream sides, and roadsides at altitudes of 750–1500 m, mixed with the white flower phenotype individuals. Precisely because red and white phenotypes individuals shared the same habitat, we ruled out the possibility of the elements of calcium and phosphorus causing the difference in flower color [62,63].

Phylogenetically, chloroplast genome and ITS trees are identical, both of which strongly supported the monophyly of the species *P. fortuneana*, and *P. fortuneana* (red) was nested within *P. fortuneana* (white) (Figure 9 and Appendix A), undoubtedly supporting that the red flower phenotype of *P. fortuneana* should belong to the species of *P. fortuneana*. Moreover, comparative chloroplast genome analyses between red and white flower phenotypes of *P. fortuneana* also revealed there is a very close genetic relationship between them. Based on above evidence, we thus suggested that the red flower phenotype of *P. fortuneana* could be recognized as a rare phenotype resource of this species, conforming to the initial view of the first discoverer [17], or be considered as a new forma.

### 4.3. Limitations and Future Directions

In conclusion, the findings obtained in this study will not only provide new insights into chloroplast genome evolution and phylogeny of *Pyracantha*, and the taxonomic status of red flower phenotype of *P. fortuneana*, but also be useful for breeding, cultivation and utilization of this economically important species. However, at least two limitations of our study should be acknowledged. First, current sampling is too sparse for the red flower phenotype of *P. fortuneana* (only one individual), thus additional sampling is needed to resolve intra-specific relationships and to evaluate how well species delimitations based primarily on morphology coincide with chloroplast genome lineages. Second, although previous studies have reported that flower color in Rosaceae is mainly attributed to anthocyanin accumulation, and controlled by the transcription factor classes of MYB, bHLH, and WD40, e.g., [64,65], the genomic/genetic data currently available for these two flower phenotypes of *P. fortuneana* are inadequate to uncover the exact molecular mechanism underlying the flower color variations in this species. Thus, the family classic genetic techniques, along with transcriptome profiling, gene expression and population genetic data and/or approaches, are needed to identify putative loss-of-function mutations and/or gene expression changes that generate rare, red flowers instead of the common, white color in *P. fortuneana*. Finally, it is also worth emphasizing that within-population flower color variation is relatively uncommon, the population of *P. fortuneana* from the Enshi Tujia and Miao Autonomous Prefecture, western Hubei, China provides an ideal system to explore the possible mechanisms that maintain flower color variation within populations, which will be useful in understanding fundamental evolutionary processes that create and maintain trait variation [66].

## Figures and Tables

**Figure 1 genes-13-02404-f001:**
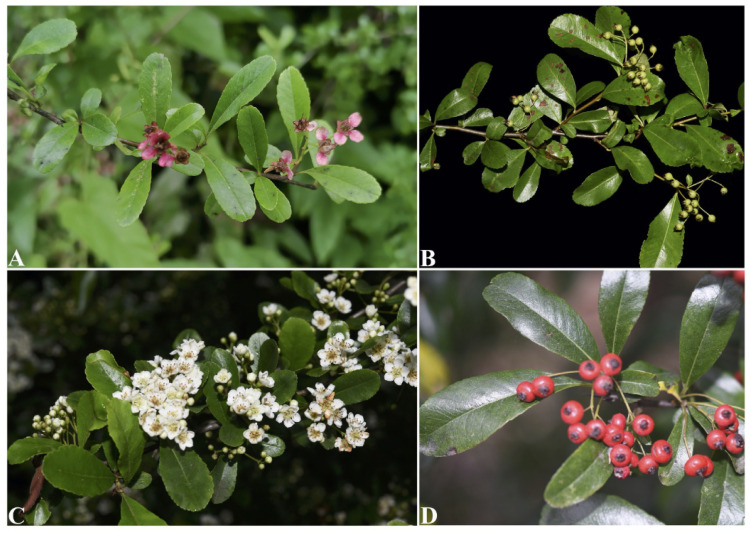
Morphological comparison between red and white flower phenotypes of *P. fortuneana*. (**A**) (**B**) the flowers and young fruits of individual with the red flower phenotype; (**C**,**D**) the flowers and mature fruits of individual with the white flower phenotype (photographed by Shi-Xiong Ding).

**Figure 2 genes-13-02404-f002:**
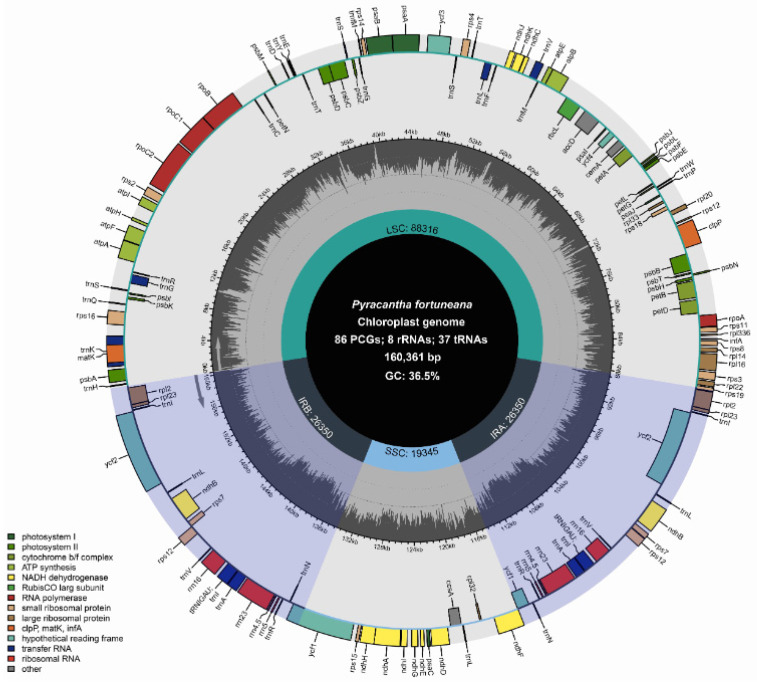
The circular chloroplast genome map of *P. fortuneana* (red). The genes drawn inside the circle are transcribed counterclockwise while those outside are transcribed clockwise. Different functional genes are differently colored on the outer circle. The dashed darker gray area in the inner circle denotes GC content, and the lighter gray area corresponds to AT content.

**Figure 3 genes-13-02404-f003:**
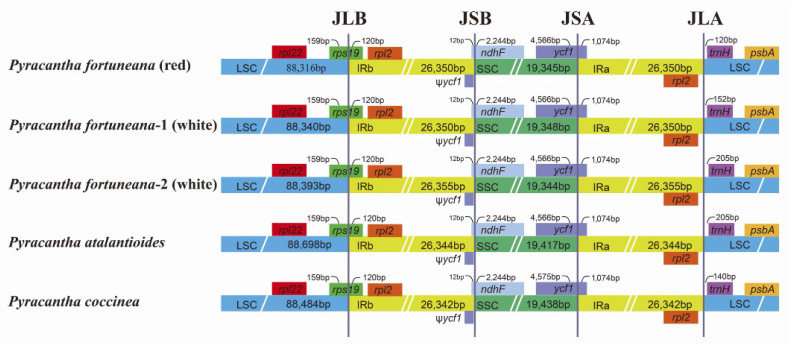
Comparative junction characteristics of LSC, SSC, and IR regions of chloroplast genomes between three accessions of *P. fortuneana* and two other *Pyracantha* species. JLB, JSB, JSA, and JLA represent four different junction regions in the chloroplast genome boundaries.

**Figure 4 genes-13-02404-f004:**
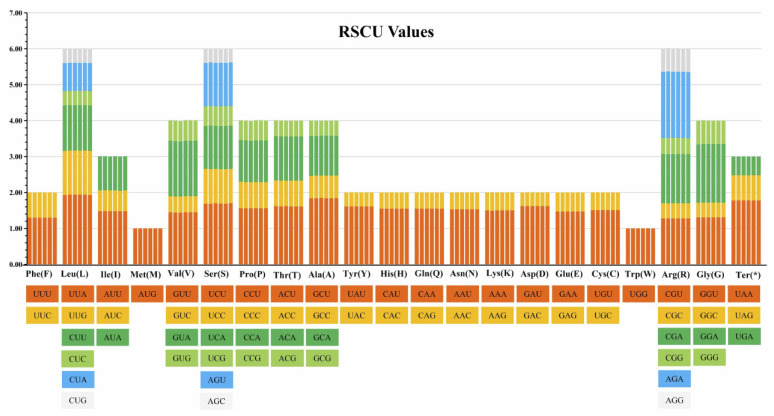
Comparative analysis plots of RSCU values for the six *Pyracantha* chloroplast genomes. Each amino acid corresponds to six histograms, and their heights represent the RSCU value. The histograms from left to right are *P. atalantioides*, *P. angustifolia*, *P. coccinea*, *P. fortuneana* (red), *P. fortuneana*-1 (white), and *P. fortuneana*-2 (white).

**Figure 5 genes-13-02404-f005:**
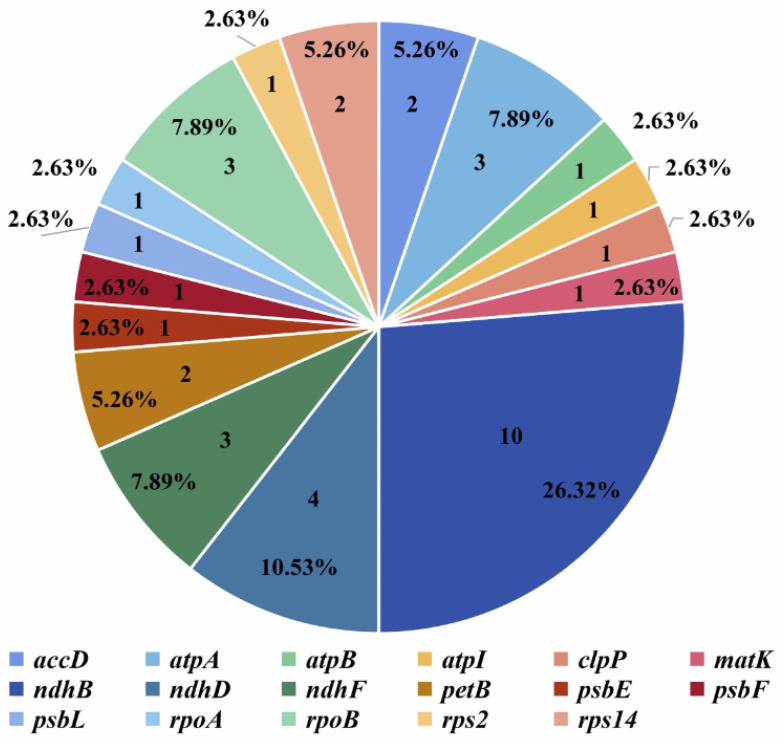
The predicted RNA editing sites in protein-coding genes of *P. fortuneana* chloroplast genomes.

**Figure 6 genes-13-02404-f006:**
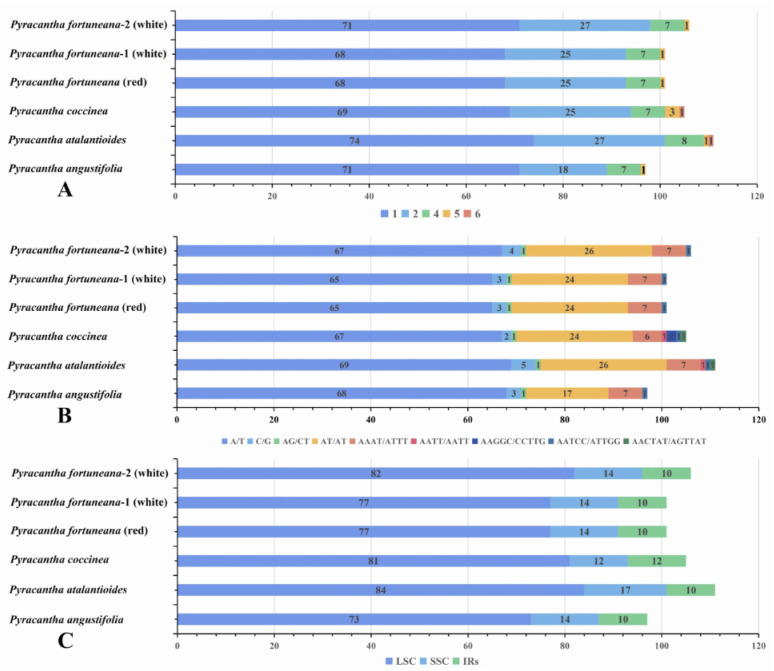
Analyses of SSRs in six *Pyracantha* chloroplast genomes. (**A**) Frequency of SSR types. (**B**) Frequency of different SSR units. (**C**) Frequency of SSRs in LSC, SSC, and IR regions.

**Figure 7 genes-13-02404-f007:**
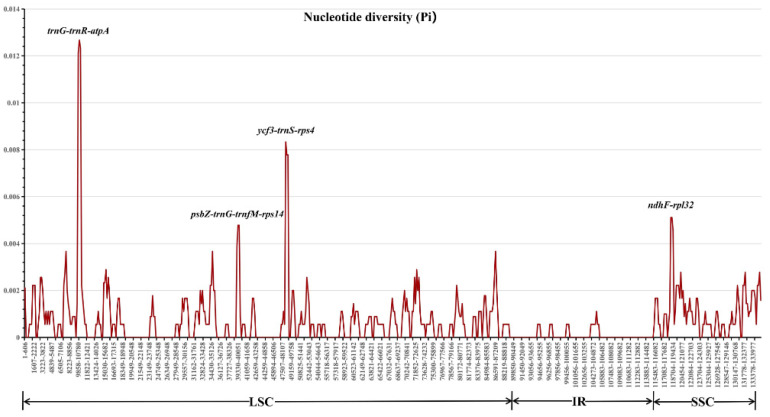
The nucleotide diversity (Pi) of six *Pyracantha* chloroplast genomes.

**Figure 8 genes-13-02404-f008:**
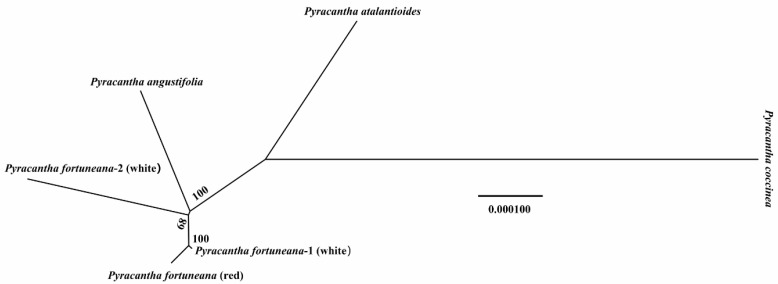
Distance-based NJ tree of the genus *Pyracantha* inferred from chloroplast genome sequences.

**Figure 9 genes-13-02404-f009:**
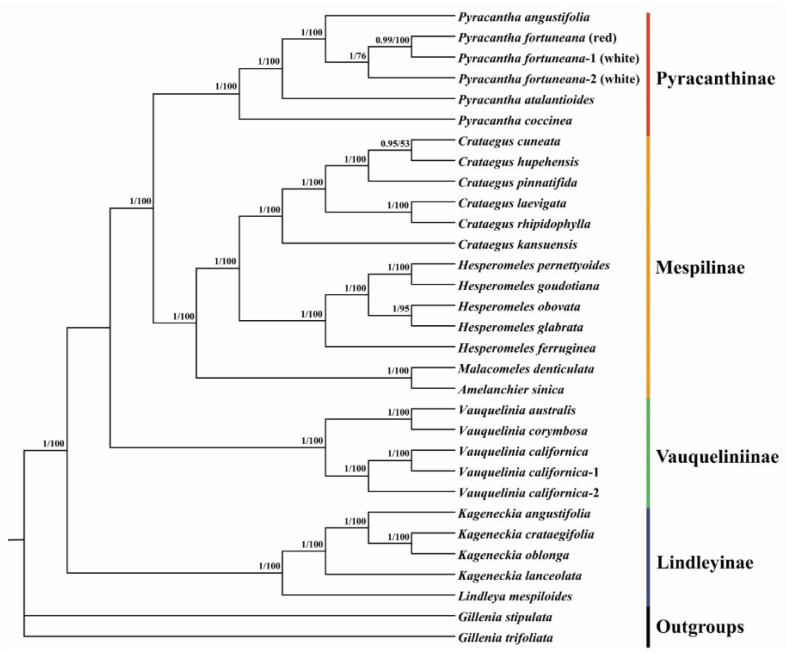
The phylogenetic tree of the genus *Pyracantha* inferred from chloroplast genome sequences based on the methods of maximum likelihood (ML) and Bayesian inference (BI). The BI posterior probabilities/ML bootstrap values are displayed above the lines.

**Table 1 genes-13-02404-t001:** The basic characteristics of five *Pyracantha* chloroplast genomes.

Species		*P. fortuneana* (red) *	*P. fortuneana*-1 (white)	*P. fortuneana*-2 (white)	*P. coccinea*	*P. atalantioides*
NCBI Accession		OM793283	NC_059101	MW596361	NC_062343	MW801001
Size (bp)	Whole	160,361	160,388	160,447	160,606	160,803
	LSC	88,316	88,340	88,393	88,484	88,698
	SSC	19,345	19,348	19,344	19,438	19,417
	IR	26,350	26,350	26,355	26,342	26,344
GC content (%)	Whole	36.5	36.5	36.5	36.5	36.4
	LSC	34.1	34.1	34.1	34.1	34
	SSC	30.4	30.4	30.4	30.3	30.4
	IR	42.7	42.7	42.7	42.7	42.7
Total genes		131	131	131	131	131
Genes (Duplicated)	PCGs	86 (7)	86 (7)	86 (7)	86 (7)	86 (7)
	tRNA	37 (7)	37 (7)	37 (7)	37 (7)	37 (7)
	rRNA	8 (4)	8 (4)	8 (4)	8 (4)	8 (4)

*, newly sequenced chloroplast genome.

**Table 2 genes-13-02404-t002:** Gene composition of the *P. fortuneana* (red) chloroplast genome.

Category	Groups of Genes	Name of Genes
**Self-replication**	Ribosomal RNA	*rrn4.5* (2×), *rrn5* (2×), *rrn16* (2×), *rrn23* (2×)
	Transfer RNA	*trnA-UGC* ^1^(2×), *trnC-GCA*, *trnD-GUC*, *trnE-UUC*, *trnF-GAA*, *trnG-GCC*, *trnG-UCC*^1^, *trnH-GUG*, *trnI-CAU* (2×), *trnI-GAU* ^1^, (2×), *trnK-UUU* ^1^, *trnL-CAA* (2×), *trnL-UAA* ^1^, *trnL-UAG*, *trnM-CAU*, *trnfM-CAU*, *trnN-GUU* (2×), *trnP-UGG*, *trnQ-UUG*, *trnR-UCU*, *trnR-ACG* (2×), *trnS-UGA*, *trnS-GCU*, *trnS-GGA*, *trnT-GGU*, *trnT-UGU*, *trnV-UAC* ^1^, *trnV-GAC* (2×), *trnW-CCA*, *trnY-GUA*
	Small subunit of ribosome	*rps2*, *rps3*, *rps4*, *rps7* (2×), *rps8*, *rps11*, *rps12* ^1,^(2×), *rps14*, *rps15*, *rps16* ^1^, *rps18*, *rps19*
	Large subunit of ribosome	*rpl2*^1,^(2×), *rpl14*, *rpl16*^1^, *rpl20*, *rpl22*, *rpl23* (2×), *rpl32*, *rpl33*, *rpl36*
	RNA polymerase subunits	*rpoA*, *rpoB*, *rpoC1*^1^, *rpoC2*
**Photosynthesis**	Photosystem I	*psaA*, *psaB*, *psaC*, *psaI*, *psaJ*
	Photosystem II	*psbA*, *psbB*, *psbC*, *psbD*, *psbE, psbF*, *psbH*, *psbI*, *psbJ*, *psbK*, *psbL*, *psbM*, *psbN*, *psbT*, *psbZ*
	Subunits of cytochrome	*petA*, *petB*^1^, *petD*^1^, *petG*, *petL*, *petN*
	ATP synthase	*atpA*, *atpB*, *atpE*, *atpF*^1^, *atpH*, *atpI*
	NADH-dehydrogenase	*ndhA*^1^, *ndhB*^1,^, *ndhC*, *ndhD*, *ndhE*, *ndhF*, *ndhG*, *ndhH*, *ndhI*, *ndhJ*, *ndhK*
**Other genes**	Rubisco large subunit	*rbcL*
	Maturase K	*matK*
	Envelope membrane protein	*cemA*
	Acetyl-CoA carboxylase	*accD*
	Proteolysis	*clpP* ^2^
	Cytochrome c biogenesis	*ccsA*
**Unknown**	Conserved open reading frames	*ycf1*, *ycf2* (2×), *ycf3* ^2^, *ycf4*, *ycf15* (2×)

^1^, genes with one intron; ^2^, genes with two introns; (2×), two genes copied in IR regions.

**Table 3 genes-13-02404-t003:** Codon usage in the chloroplast genome of *P. fortuneana* (red).

Amino Acids	Codons	Count	RSCU	Amino Acids	Codons	Count	RSCU
Leucine	UUA(L)	891	1.93	Phenylalanine	UUU(F)	966	1.3
	UUG(L)	568	1.23		UUC(F)	521	0.7
	CUU(L)	580	1.26	Tyrosine	UAU(Y)	790	1.61
	CUC(L)	184	0.4		UAC(Y)	191	0.39
	CUA(L)	359	0.78	Histidine	CAU(H)	491	1.55
	CUG(L)	181	0.39		CAC(H)	141	0.45
Isoleucine	AUU(I)	1113	1.48	Glutamine	CAA(Q)	729	1.55
	AUC(I)	433	0.58		CAG(Q)	213	0.45
	AUA(I)	711	0.95	Asparagine	AAU(N)	973	1.53
Methionine	AUG(M)	622	1		AAC(N)	300	0.47
Valine	GUU(V)	514	1.45	Lysine	AAA(K)	1049	1.5
	GUC(V)	159	0.45		AAG(K)	353	0.5
	GUA(V)	547	1.54	Aspartic Acid	GAU(D)	881	1.62
	GUG(V)	202	0.57		GAC(D)	207	0.38
Serine	UCU(S)	572	1.7	Glutamic Acid	GAA(E)	1014	1.47
	UCC(S)	322	0.96		GAG(E)	361	0.53
	UCA(S)	405	1.2	Cysteine	UGU(C)	226	1.51
	UCG(S)	186	0.55		UGC(C)	74	0.49
	AGU(S)	408	1.21	Tryptophan	UGG(W)	452	1
	AGC(S)	129	0.38	Arginine	CGU(R)	339	1.28
Proline	CCU(P)	420	1.56		CGC(R)	111	0.42
	CCC(P)	195	0.73		CGA(R)	363	1.37
	CCA(P)	311	1.16		CGG(R)	118	0.44
	CCG(P)	149	0.55		AGA(R)	490	1.84
Threonine	ACU(T)	544	1.61		AGG(R)	173	0.65
	ACC(T)	243	0.72	Glycine	GGU(G)	582	1.31
	ACA(T)	416	1.23		GGC(G)	181	0.41
	ACG(T)	149	0.44		GGA(G)	721	1.63
Alanine	GCU(A)	642	1.84		GGG(G)	290	0.65
	GCC(A)	218	0.63	Stop codon	UAA(*)	51	1.78
	GCA(A)	385	1.11		UAG(*)	20	0.7
	GCG(A)	148	0.42		UGA(*)	15	0.52

**Table 4 genes-13-02404-t004:** The purifying selected sites detected in the 78 CDSs of six *Pyracantha* chloroplast genomes.

Codon	Gene	Region	Synonymous Substitution Rate	Non-Synonymous Substitution Rate	*p*-Value	Total Branch Length
Y	*atpI*	LSC	4368.606	0.067	0.0694	0.401
N	*ccsA*	SSC	4368.606	0.067	0.0403	0.238
A	*matK*	LSC	680.592	0.068	0.099	0.300
P	*ndhD*	SSC	628.859	0.069	0.0874	0.210
P			628.859	0.069	0.0874	0.210
K	*ndhF*	SSC	1327.452	0.070	0.0565	0.376
A	*ndhH*	SSC	1526.202	0.068	0.0352	0.500
Y	*psbM*	LSC	1687.808	0	0.0738	0.448
G	*rbcL*	LSC	970.757	0	0.0953	0.288
P	*rpoC1*	LSC	1822.725	0	0.0715	0.317
K	*rps18*	LSC	2081.359	0	0.0538	0.339
T			1168.659	0	0.0942	0.348

**Table 5 genes-13-02404-t005:** The genetic distance among six *Pyracantha* chloroplast genomes.

	Species	1	2	3	4	5	6
1	*P. fortuneana* (red)						
2	*P. fortuneana*-1 (white)	0.000					
3	*P. fortuneana*-2 (white)	0.000	0.000				
4	*P. angustifolia*	0.000	0.000	0.000			
5	*P. atalantioides*	0.000	0.000	0.001	0.001		
6	*P. coccinea*	0.001	0.001	0.001	0.001	0.001	

**Table 6 genes-13-02404-t006:** Morphological, habit, and habitat differences between *P. fortuneana* individuals with red and white flower phenotypes.

Characters	Red Flower Phenotype	White Flower Phenotype
Plant habit	evergreen or semi-evergreen shrub, mature leaves turn red in winter	evergreen shrub, mature leaves keep green in winter
Leaf	leaf blade obovate to obovate-oblong, base cuneate to wide round	leaf blade obovate-oblong, base cuneate, extending down to the petiole
Serrations	serrations conspicuous or inconspicuous	serrations obtuse or inconspicuous
Inflorescence	compound corymb, loose, fewer flowers	compound corymb, dense, more flowers
Calyx lobes	inner sepals red to pink	inner sepals white
Flower	petals, styles and filaments red to pink	petals, styles and filaments white
Phenology	fl. Mar–May, fr. Aug–Nov	fl. Mar–June, fr. Aug–Nov
Habitat	Alt. 750–1500 m	Alt. 500–2800 m

## Data Availability

The chloroplast genome sequence and rDNA ITS sequence of P. fortuneana (red) can be found in the GenBank (https://www.ncbi.nlm.nih.gov/genbank/) under accession numbers OM793283 and OP821228, respectively. Raw Illumina data were deposited at NCBI SRA (no. SRR21976475).

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
