# Peer review of "Morphological Characteristics and Comparative Chloroplast Genome Analyses between Red and White Flower Phenotypes of Pyracantha fortuneana (Maxim.) Li (Rosaceae), with Implications for Taxonomy and Phylogeny"

_genes, 2022, doi:10.3390/genes13122404_

Round 1
Reviewer 1 Report
Dear authors; Your article titled “Morphological Characteristics and Comparative Chloroplast Genome Analyses between Red and White Flower Phenotypes of Pyracantha fortuneana (Maxim.) Li (Rosaceae), with implications for taxonomy and phylogeny” is an original and high quality work. However, some major revisions need to be made on your article. I recommend that you carefully review the revision notes I have suggested below.
Introduction
I suggested that the authors better added their hypothesis before the aim study in latest paragraph. Moreover, to verify this hypothesis, mention to the utilized parameters.
Results
The analyses of the molecular data are restricted to a Phylogenetic relationship. However, a more rigorous data analyses should have been performed. The following analyses are suggested to be performed and thoroughly discussed;
a. Principal Component analyses to visualize the genetic relationship among taxa evaluated
b. AMOVA analyses to deduce the partition of the genetic variance within and among species
c. The morphological data from the taxa to provide the perspective
Discussion
The Discussion needs to emphasize the results and to explain more clearly how they represent an advance on previous studies calcium and phosphorus involving mitigate phytotoxicity with your aim.
Dear Author more emphasis should be placed on morphological characters in the discussion, Important morphological characters of taxa can be presented in a table to facilitate differentiation
General Comment
Please removed all the older references in the all-manuscript section.
Reviewer 2 Report
It was a good study to confirm the difference in chromosomal DNA of red flowering pyracantha with high ornamental value. However, as a result, it was regrettable that the difference according to phenotype could not be found in chloroplast genome. Have you ever studied any comparative studies on mtDNA?
For the manuscript, there was no particular modification except a small part in figure 2. The gene transcription direction seems to be marked differently from the figure legend. Please check it out.
Round 2
Reviewer 1 Report
Dear author, thank you for developing the article taking into account our suggestions. However, I recommend you to use the Principal Component analyzes and AMOVA analyzes that I recommend in this study. Considering the current corrections you have made, I suggest that the manuscript be accepted.
Sincerely